# Water Requirement in North China from Grey Point Prediction and Grey Interval Prediction



**Lifeng Wu** [1,2,*] and **Yuan Meng** [1,2,3]

1   School of Water Conservancy and Hydroelectric Power, Hebei University of Engineering,
    Handan 056038, China
2   Hebei Key Laboratory of Intelligent Water Conservancy, Hebei University of Engineering,
    Handan 056038, China
3   School of Materials Science and Engineering, Hebei University of Engineering, Handan 056038, China
*   Correspondence: wulifeng@hebeu.edu.cn

**Abstract:** Since the implementation of the sustainable development strategy, China has made great efforts to save water resources. Therefore, effective prediction and analysis of regional water consumption are very important for the regional economy. In order to forecast the water requirement of the five provinces in North China, the DGMC(1,2) model is proposed to predict the point value of water requirement by considering the three industries and the population. The results turn out that DGMC(1,2) model is an efficient way of predicting water requirements. In addition, the interval value of water requirement is predicted by the establishment of the interval DGMC(1,2) model. According to the prediction results, the variation trend of water requirement in each region is analyzed in detail, and the corresponding suggestions are put forward. The results can have practical value and be used for policy-making.

**Keywords:** water requirement; point prediction; interval prediction; population; influence factor





## 1. Introduction

Since the implementation of the sustainable development strategy, China government has made great efforts to save water resources. However, water resources still cannot meet regional economic development. Therefore, the effective prediction and analysis of regional water requirements are very important for the development of the regional economy. The degree of accuracy of prediction results straightly affects the effect of the policy-making for water requirement scheduling [1].

For the present, the prediction methods of water requirement are various, including Artificial Neural Networks [2–5], Fuzzy and neuro-fuzzy models [6,7], Support Vector Machine [8,9], Markov process [10,11], data assimilation technique [12], Kalman filter [13], Regression forecasting model [14], Metaheuristics [15], Moving Window [16], and System dynamics [17,18], statistical analysis [19–22]. The foregoing studies have applied all sorts of methods and their hybrids to forecast the water requirement. Nevertheless, the above-mentioned models are mainly based on big data. Considering some problems of water demand prediction, it is hard to achieve plenty of original water requirement data and to cater to the data needs of statistical analysis and neural networks. Meanwhile, considering the practical and operable applicability, the grey prediction method is superior remarkably, which is proposed by Professor Deng Julong to settle the problem of uncertain data or limited samples [23]. Currently, the grey prediction model has drawn the attention of a large number of people and is usually used in short-period or medium-period prediction because of its remarkable effect [24–26].

In addition, the grey system theory has also been used extensively in the study of water requirement prediction. Xu et al. used the fractional-order cumulative, discrete grey model to forecast agricultural water demand in two regions of China. The results indicate that the

model has superior prediction properties than the grey model with one variable [27]. Wu et al. established a new greywater requirement prediction model, explored the parameter estimation and error testing methods on it, and used this model to predict the water consumption in Chongqing [28]. Besides, Qiao et al. proposed a fractional cumulative grey forecasting model (FGM(1,1)) to predict water demand in various regions of China [29]. Since the traditional grey multivariate model does not satisfy the new information priority principle, and the accumulative operator can change the nature of the model [30], we make a point prediction of water requirement in the five provinces of North China based on the grey multivariate model (DGMC(1,2)) from Reference [30]. We also proposed an interval DGMC(1,2) to make the interval prediction of water requirement. The corresponding measures to the variation of water requirement under different influence factors are put forward.

The structure of this paper is organized as follows: the research area and data are discussed in Section 2. The results of point prediction in North China from 2022 to 2026 are stated in Section 3. The results of interval prediction are listed in Section 4. The discussion and conclusion are offered in the final Section.

## 2. Research Area and Data

### 2.1. Research Area

Since the altitude of North China is mostly below 50 m, the area is characterized by a temperate monsoon climate. Compared with most areas in China, where the precipitation is concentrated from May to October, the precipitation in North China is mainly concentrated in July and August, which can reach 80% of the annual precipitation. Water resources are unevenly distributed in North China.

This region is characterized by high population densities and the largest comprehensive industrial base with rich resources. However, North China is the most typical resource-based water shortage area in China. The water resource per capita in the Beijing-Tianjin-Hebei region is only 1/9 of that of the whole country. With 2.3% of the national land area and less than 1% of the water resource, this region carries 8.0% of the national population, 9.5% of the industrial added value, 10.1% of the GDP, and nearly 30% of the steel output. The lack of water resources is the most unfavorable factor for economic development in North China. Thus, it is more beneficial for forecasting the water requirement in this region for the coordinated and sustainable development of the regional economy. The cities of Hebei, Shanxi, Inner Mongolia autonomous region, Tianjin, and Beijing are selected as research areas, and their geographic locations are presented in Figure 1. We will study these five places respectively and then make a total analysis of North China.

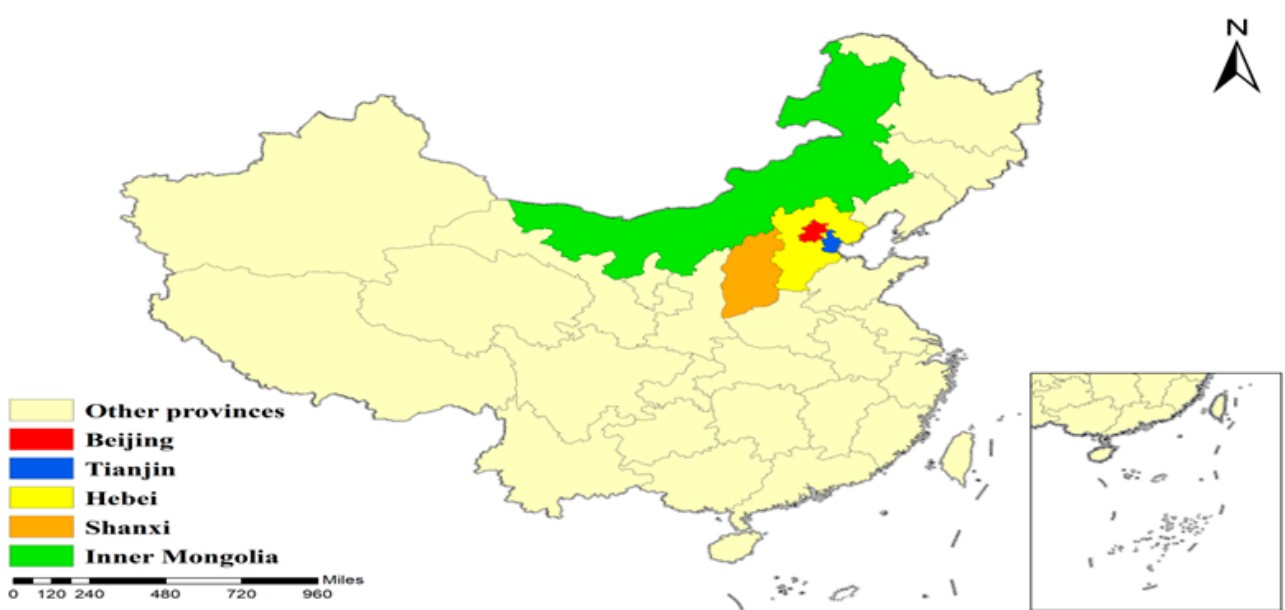

**Figure 1.** Geographical location of the research area.

*2.2. Data Source*

By predicting the water requirement with the socioeconomic indicators in this region, the pressure of lack of water resources can be effectively alleviated. Therefore, the water requirement from 2016 to 2021 is selected as the research sample in this part. According to the relevant literature, the added value of the primary industry (AVPI), the secondary industry (AVSI), the tertiary industry (AVTI), and the population of the year-end (YEP) of five provinces in north China are taken as the influence factors of the water requirement. The relevant indicators are shown in Table 1.

**Table 1.** Social and economic index.

| Index | Unit |
|---|---|
| AVPI | 100 million Yuan |
| AVSI | 100 million Yuan |
| AVTI | 100 million Yuan |
| YEP | 10,000 people |
| Water requirement | 100 million m$^3$ |

Four socioeconomic indicators and the water requirement in the five provinces of North China are obtained from the China County Statistical Yearbook and China Water Resources Bulletin from 2016 to 2021. The related data are shown in Tables 2–6.

**Table 2.** The related data in Beijing.

| Year | AVPI | AVSI | AVTI | YEP | Water Requirement |
|---|---|---|---|---|---|
| 2016 | 129.79 | 4944.44 | 20,594.9 | 2195 | 38.8 |
| 2017 | 120.42 | 5326.76 | 22,567.76 | 2194 | 39.5 |
| 2018 | 118.69 | 5647.65 | 24,553.64 | 2192 | 39.3 |
| 2019 | 113.69 | 5715.06 | 29,542.53 | 2190 | 41.7 |
| 2020 | 107.61 | 5716.37 | 30,278.57 | 2189 | 40.6 |
| 2021 | 111.34 | 7268.60 | 32,889.61 | 2189 | 40.8 |

**Table 3.** The related data in Tianjin.

| Year | AVPI | AVSI | AVTI | YEP | Water Requirement |
|------|------|------|------|-----|-------------------|
| 2016 | 220.22 | 7571.35 | 10,093.82 | 1443 | 27.2 |
| 2017 | 168.96 | 7593.59 | 10,786.64 | 1410 | 27.5 |
| 2018 | 172.71 | 7609.81 | 11,027.12 | 1383 | 28.4 |
| 2019 | 185.23 | 4969.18 | 8949.87 | 1385 | 28.4 |
| 2020 | 210.18 | 4804.08 | 9069.47 | 1387 | 27.8 |
| 2021 | 225.41 | 5854.27 | 9615.37 | 1373 | 32.3 |

**Table 4.** Related data in Hebei Province.

| Year | AVPI | AVSI | AVTI | YEP | Water Requirement |
|------|------|------|------|-----|-------------------|
| 2016 | 3492.81 | 15,256.93 | 13,320.71 | 7375 | 182.6 |
| 2017 | 3129.98 | 15,846.21 | 15,040.13 | 7409 | 181.6 |
| 2018 | 3338 | 16,040.06 | 16,632.21 | 7426 | 182.4 |
| 2019 | 3518.44 | 13,597.26 | 17,988.82 | 7447 | 182.3 |
| 2020 | 3880.14 | 13,597.2 | 18,729.54 | 7464 | 182.8 |
| 2021 | 4030.3 | 16,364.2 | 19,996.7 | 7448 | 181.9 |

**Table 5.** The related data in Shanxi Province.

| Year | AVPI | AVSI | AVTI | YEP | Water Requirement |
|------|------|------|------|-----|-------------------|
| 2016 | 784.78 | 5028.99 | 7236.64 | 3514 | 75.5 |
| 2017 | 719.16 | 6778.89 | 8030.37 | 3510 | 74.9 |
| 2018 | 760.64 | 7089.19 | 8988.28 | 3502 | 74.3 |
| 2019 | 824.72 | 7453.09 | 8748.87 | 3497 | 76.0 |
| 2020 | 946.68 | 7675.44 | 9029.81 | 3490 | 72.8 |
| 2021 | 1286.9 | 11,213.1 | 10,090.2 | 3480 | 72.6 |

**Table 6.** The related data in Inner Mongolia autonomous region.

| Year | AVPI | AVSI | AVTI | YEP | Water Requirement |
|------|------|------|------|-----|-------------------|
| 2016 | 1637.39 | 5579.77 | 7937.08 | 2436 | 190.3 |
| 2017 | 1649.77 | 5874.25 | 8046.76 | 2433 | 188.0 |
| 2018 | 1753.82 | 6335.38 | 8728.10 | 2422 | 192.1 |
| 2019 | 1863.19 | 6763.14 | 8728.1 | 2415 | 190.9 |
| 2020 | 2025.12 | 6908.17 | 8466.66 | 2403 | 194.4 |
| 2021 | 2225.2 | 9374.19 | 8914.8 | 2400 | 191.7 |

## 3. Grey Point Prediction of Water Requirement in North China

Traditional econometric forecasting methods on water requirements tend to require more data and do not value new data, so the prediction effect is not good. Grey multivariable model not only requires fewer data and attaches importance to new data but also can make more accurate judgments on the complex system and provide valuable information for the decision maker. The cost of the grey multivariable model is small. Therefore we choose a grey multivariable model. In this section, the process modeling of DGMC(1,$N$) is listed.

Step 1: Assume the non-negative sequence $X_1^{(0)} = \left\{ x_1^{(0)}(1), x_1^{(0)}(2), \cdots, x_1^{(0)}(m) \right\}$ as the system characteristic sequence, and the corresponding correlation factor sequences are

$$X_2^{(0)} = \left\{ x_1^{(0)}(1), x_1^{(0)}(2), \cdots, x_1^{(0)}(m) \right\},$$
$$X_3^{(0)} = \left\{ x_1^{(0)}(1), x_1^{(0)}(2), \cdots, x_1^{(0)}(m) \right\},$$
$$\cdots$$
$$X_N^{(0)} = \left\{ x_1^{(0)}(1), x_1^{(0)}(2), \cdots, x_1^{(0)}(m) \right\}$$

Step 2: Taking the 0.1-order accumulation sequence as an example, the DGMC(1,*N*) model is shown in Equation (1)

$$X_1^{(0)}(k) + b_1 z_1^{(0.1)}(k) = \sum_{i=2}^{N} b_1 z_1^{(0.1)}(k) + u \tag{1}$$

$b_1, b_2, \cdots, b_N$ and $u$ are parameters to be estimated. The least squares method is used to minimize the residual sum of squares. The unknown parameters are solved by Equation (2)

$$\left[ \hat{b}_1, \hat{b}_2, \cdots, \hat{b}_u, \hat{u} \right] = (B^T B)^{-1} B^T Y \tag{2}$$

where,

$$B = \begin{bmatrix} -\frac{x_1^{(0.1)}(1)+x_1^{(0.1)}(2)}{2} & -\frac{x_2^{(0.1)}(1)+x_2^{(0.1)}(2)}{2} & \cdots & -\frac{x_N^{(0.1)}(1)+x_N^{(0.1)}(2)}{2} & 1 \\ -\frac{x_1^{(0.1)}(2)+x_1^{(0.1)}(3)}{2} & -\frac{x_2^{(0.1)}(2)+x_2^{(0.1)}(3)}{2} & \cdots & -\frac{x_N^{(0.1)}(2)+x_N^{(0.1)}(3)}{2} & 1 \\ -\frac{x_1^{(0.1)}(3)+x_1^{(0.1)}(4)}{2} & -\frac{x_2^{(0.1)}(3)+x_2^{(0.1)}(4)}{2} & \cdots & -\frac{x_N^{(0.1)}(3)+x_N^{(0.1)}(4)}{2} & 1 \\ \vdots & \vdots & \vdots & \vdots & \vdots \\ -\frac{x_1^{(0.1)}(m-1)+x_1^{(0.1)}(m)}{2} & -\frac{x_2^{(0.1)}(m-1)+x_2^{(0.1)}(m)}{2} & \cdots & -\frac{x_N^{(0.1)}(m-1)+x_N^{(0.1)}(m)}{2} & 1 \end{bmatrix},$$

$$Y = \begin{bmatrix} x_1^{(0.1)}(2) - x_1^{(0.1)}(1) \\ x_1^{(0.1)}(3) - x_1^{(0.1)}(2) \\ x_1^{(0.1)}(4) - x_1^{(0.1)}(3) \\ \vdots \\ x_1^{(0.1)}(m) - x_1^{(0.1)}(m-1) \end{bmatrix}.$$

Step 3: Assume $\hat{x}_1^{(1)}(1) = x_1^{(0)}(1)$, then

$$\hat{x}_1^{(0.1)}(k) = x_1^{(0)} e^{-\hat{b}_1(k-1)} + \int_1^k e^{-\hat{b}_1(k-1)} f(t) dt \tag{3}$$

where $f(t) = b_2 x_2^{(0.1)}(t) + b_3 x_3^{(0.1)}(t) + \cdots + b_N x_N^{(0.1)}(t) + u$.

The time response equation gained from the Gaussian formula is

$$\hat{x}_1^{(0.1)}(k) = x_1^{(0)}(k) e^{-\hat{b}_1(k-1)} + \sum_{t=2}^{k} \left\{ e^{-\hat{b}_1(k-t+\frac{1}{2})} * \frac{1}{2} [f(t) + f(t-1)] \right\}. \tag{4}$$

Step 4: The fitting sequence is

$$\hat{x}_1^{(0)} = \left\{ \hat{x}_1^{(0)}(1), \hat{x}_1^{(0)}(2), \cdots, \hat{x}_1^{(0)}(m) \right\},$$

where $\hat{x}_1^{(0)}(k) = \hat{x}_1^{(0.1)}(k) - 0.1\hat{x}_1^{(1)}(k-1)$.

Step 5: The model is evaluated by mean absolute percentage error (MAPE = $\frac{100\%}{m} \sum_{k=1}^{m} \left| \frac{x_1^{(0)} - \hat{x}_1^{(0)}}{x_1^{(0)}} \right|$). If the MAPE is less than 10%, the model is judged to fit well. It is assumed that the correlation factor sequences of the prediction period have been obtained. By the 0.1-order accumulation generator, the accumulated correlation factor sequences

are substituted into Equation (4) to obtain the system feature sequence of the prediction period, denoted as $y_{0.1}(m+1), y_{0.1}(m+2), \cdots$. The point value of the system characteristic sequence at other time points is obtained. The results of point prediction in North China from 2022 to 2026 are listed in this Section.

### 3.1. Validating the Model of Water Requirement in Beijing

To validate the model, due to the impact of 19-COVID, because the real values for 2022 are not released by now, the data from 2016 to 2020 is the sample. The DGMC(1,2) is used for predicting the water requirement in 2021. AVPI in Beijing is the influence factor of water requirement. In order to obtain the minimum MAPE, the optimal order is solved by the particle swarm optimization algorithm. The comparison data between the fitted value and the real value is presented in Table 7. The MAPE of DGMC(1,2) in the sample data is 1.5%, and the MAPE of DGMC(1,2) in the out-of-sample data is 3.7%. Both the fitted MAPE and the predicted MAPE are far less than 10%. DGMC(1,2) error is small. It indicates that the model can obtain the better predictive result of the water requirement in Beijing.

**Table 7.** The fitted value and the real value of water requirement.

| Year | The Real Value | | The Fitted Value |
|---|---|---|---|
| 2016 | 38.8 | | 38.8 |
| 2017 | 39.5 | | 39.1 |
| 2018 | 39.3 | | 40.0 |
| 2019 | 41.7 | | 40.7 |
| 2020 | 40.6 | | 41.5 |
| MAPE | | 1.5% | |
| 2021 | 40.8 | | 42.3 |
| MAPE | | 3.7% | |

### 3.2. Predictive Modeling of Water Requirement in Beijing

The data from 2016 to 2021 are the sample. The DGMC(1,2) is used for predicting the water requirement. AVPI in Beijing is the influence factor of water requirement. In order to obtain the minimum MAPE, the optimal order is solved by the particle swarm optimization algorithm. Judging from the change of AVPI in Beijing in the past six years, it is not difficult to see that from 2016 to 2021, AVPI in Beijing showed a stable decline with an annual mean growth rate of −3%. Therefore, it is assumed that the AVSI, AVTI, and YEP of Beijing will be unchanged in the future five years, and the growth rate of AVPI will be −2% and 1%, respectively. The predictive results of water requirement with different growth rates of AVPI in Beijing can be seen in Table 8.

**Table 8.** Prediction results of water requirement with different growth rates of AVPI in Beijing.

| Year | 2022 | 2023 | 2024 | 2025 | 2026 |
|---|---|---|---|---|---|
| −2% | 40.47 | 40.10 | 39.39 | 38.30 | 36.77 |
| 1% | 40.97 | 41.27 | 41.70 | 42.25 | 42.94 |

Similarly, taking AVSI, AVTI, and YEP as the influence factors, the DGMC(1,2) model is established. The calculative process was consistent with the above-mentioned. Based on the influence factor of the recent five years, we can obtain the prediction results of water requirements with different growth rates. The forecasting results are shown in Tables 9–11.

**Table 9.** Prediction results of water requirement with different growth rates of AVSI in Beijing.

| Year | 2022 | 2023 | 2024 | 2025 | 2026 |
|---|---|---|---|---|---|
| −1% | 40.14 | 39.63 | 39.24 | 38.96 | 38.76 |
| 3% | 41.07 | 41.26 | 41.41 | 41.54 | 41.64 |

**Table 10.** Prediction results of water requirement with different growth rates of AVTI in Beijing.

| Year | 2022 | 2023 | 2024 | 2025 | 2026 |
|------|------|------|------|------|------|
| −1% | 40.23 | 39.84 | 39.46 | 39.06 | 38.66 |
| 9% | 38.94 | 39.11 | 39.30 | 39.50 | 39.73 |

**Table 11.** Prediction results of water requirement with different growth rates of YEP in Beijing.

| Year | 2022 | 2023 | 2024 | 2025 | 2026 |
|------|------|------|------|------|------|
| −1% | 40.47 | 40.20 | 39.74 | 39.17 | 38.53 |
| 1% | 40.71 | 41.08 | 41.52 | 42.01 | 42.52 |

*3.3. Predictive Modeling of Water Requirement in Tianjin, Hebei, Shanxi, and Inner Mongolia Autonomous*

In this part, by using AVPI, AVSI, AVTI, and YEP as the influence factors, the DGMC(1,2) model is employed to forecast the water requirement in Tianjin (Tables 12–15), Hebei Province (Tables 16–19), Shanxi Province (Tables 20–23), Inner Mongolia autonomous region (Tables 24–27). The calculation process is in line with that of Beijing. Based on the average growth rate of each relevant factor in the last 6 years, the coming growth rate is assumed. The water requirements with the different growth rates of the relevant factors are predicted respectively.

**Table 12.** Prediction result of water requirement with different growth rates of AVPI in Tianjin.

| Year | 2022 | 2023 | 2024 | 2025 | 2026 |
|------|------|------|------|------|------|
| 1% | 30.13 | 30.53 | 30.77 | 30.97 | 31.16 |
| 2% | 30.82 | 31.47 | 31.96 | 32.40 | 32.82 |

**Table 13.** Prediction result of water requirement with different growth rates of AVSI in Tianjin.

| Year | 2022 | 2023 | 2024 | 2025 | 2026 |
|------|------|------|------|------|------|
| −6% | 24.43 | 23.60 | 22.45 | 20.78 | 18.26 |
| −4% | 24.99 | 24.43 | 23.66 | 22.55 | 20.90 |

**Table 14.** Prediction result of water requirement with different growth rates of AVTI in Tianjin.

| Year | 2022 | 2023 | 2024 | 2025 | 2026 |
|------|------|------|------|------|------|
| −2% | 25.15 | 24.75 | 24.30 | 23.74 | 23.02 |
| 1% | 31.87 | 32.34 | 32.63 | 32.77 | 32.78 |

**Table 15.** Prediction result of water requirement with different growth rates of YEP in Tianjin.

| Year | 2022 | 2023 | 2024 | 2025 | 2026 |
|------|------|------|------|------|------|
| −1% | 12.13 | 13.14 | 14.14 | 15.13 | 16.11 |
| 1% | 33.40 | 37.63 | 41.09 | 43.13 | 42.58 |

**Table 16.** Prediction result of water requirement with different growth rates of AVPI in Hebei Province.

| Year | 2022 | 2023 | 2024 | 2025 | 2026 |
|------|------|------|------|------|------|
| −1% | 178.47 | 177.00 | 175.28 | 173.24 | 170.80 |
| 1% | 187.16 | 188.46 | 190.21 | 192.60 | 195.90 |

**Table 17.** Prediction result of water requirement with different growth rates of AVSI in Hebei Province.

| Year | 2022 | 2023 | 2024 | 2025 | 2026 |
|------|------|------|------|------|------|
| −1% | 177.85 | 176.92 | 175.43 | 173.23 | 170.14 |
| 1% | 182.17 | 182.86 | 183.69 | 184.70 | 185.91 |

**Table 18.** Prediction result of water requirement with different growth rates of AVTI in Hebei Province.

| Year | 2022 | 2023 | 2024 | 2025 | 2026 |
|------|------|------|------|------|------|
| 7% | 199.65 | 207.64 | 219.16 | 236.13 | 261.55 |
| 9% | 204.43 | 215.53 | 231.95 | 256.73 | 294.72 |

**Table 19.** Prediction result of water requirement with different growth rates of YEP in Hebei Province.

| Year | 2022 | 2023 | 2024 | 2025 | 2026 |
|------|------|------|------|------|------|
| 1% | 191.82 | 195.23 | 197.67 | 198.40 | 196.12 |
| 2% | 200.20 | 207.13 | 212.64 | 215.33 | 212.63 |

**Table 20.** Prediction result of water requirement with different growth rates of AVPI in Shanxi Province.

| Year | 2022 | 2023 | 2024 | 2025 | 2026 |
|------|------|------|------|------|------|
| −1% | 69.67 | 68.74 | 68.27 | 68.06 | 68.00 |
| 5% | 79.19 | 82.75 | 88.68 | 98.53 | 114.86 |

**Table 21.** Prediction result of water requirement with different growth rates of AVSI in Shanxi Province.

| Year | 2022 | 2023 | 2024 | 2025 | 2026 |
|------|------|------|------|------|------|
| −1% | 68.93 | 68.05 | 67.71 | 67.62 | 67.66 |
| 10% | 80.77 | 86.16 | 95.51 | 111.65 | 139.36 |

**Table 22.** Prediction result of water requirement with different growth rates of AVTI in Shanxi Province.

| Year | 2022 | 2023 | 2024 | 2025 | 2026 |
|------|------|------|------|------|------|
| −1% | 70.33 | 69.36 | 68.88 | 68.71 | 68.75 |
| 5% | 77.53 | 79.90 | 83.87 | 90.43 | 101.19 |

**Table 23.** Prediction result of water requirement with different growth rates of YEP in Shanxi Province.

| Year | 2022 | 2023 | 2024 | 2025 | 2026 |
|------|------|------|------|------|------|
| −1% | 68.58 | 66.02 | 61.96 | 54.68 | 40.45 |
| 1% | 75.70 | 76.87 | 77.86 | 78.43 | 78.08 |

**Table 24.** Prediction result of water requirement with different growth rates of AVPI in Inner Mongolia.

| Year | 2022 | 2023 | 2024 | 2025 | 2026 |
|------|------|------|------|------|------|
| −1% | 190.02 | 188.25 | 186.69 | 185.33 | 184.18 |
| 5% | 210.65 | 214.97 | 219.65 | 224.81 | 230.64 |

**Table 25.** Prediction result of water requirement with different growth rates of AVSI in Inner Mongolia.

| Year | 2022 | 2023 | 2024 | 2025 | 2026 |
|------|------|------|------|------|------|
| −2% | 185.90 | 180.65 | 175.93 | 171.74 | 168.05 |
| −1% | 187.55 | 183.82 | 180.50 | 177.56 | 174.97 |

**Table 26.** Prediction result of water requirement with different growth rates of AVTI in Inner Mongolia.

| Year | 2022 | 2023 | 2024 | 2025 | 2026 |
|------|------|------|------|------|------|
| 1% | 210.85 | 214.40 | 217.97 | 221.53 | 225.09 |
| 3% | 210.47 | 216.93 | 224.74 | 234.65 | 247.80 |

**Table 27.** Prediction result of water requirement with different growth rates of YEP in Inner Mongolia.

| Year | 2022 | 2023 | 2024 | 2025 | 2026 |
|------|------|------|------|------|------|
| −1% | 188.33 | 186.84 | 184.83 | 182.55 | 180.13 |
| 1% | 189.73 | 191.43 | 193.70 | 196.29 | 199.09 |

*3.4. Results Analysis*

3.4.1. The Changing Trend of Water Requirement in Beijing

Different socioeconomic indicators in Beijing have different effects on water requirements. It can be known from Tables 8–11 when the growth rates of AVPI are −2% and 1%, AVSI is −1% and 3%, AVTI are −1% and 9%, YEP are −1% and 1%, the water requirement will all decrease or increase, and the corresponding total water requirement will change from 3.677 billion m$^3$ to 4.294 billion m$^3$ by 2026. The corresponding change will account for 9.87% and 5.24% of that in 2021 when AVPI is at the speed of −2% and 1%, respectively. Similarly, the corresponding water requirement is 3.876 billion m$^3$ and 4.164 billion m$^3$ when AVSI at the speed of −1% and 3%, respectively. The corresponding water requirement is 3.866 billion m$^3$ and 3.973 billion m$^3$ when AVTI is at the speed of −1% and 9%, respectively. The corresponding water requirement is 3.853 billion m$^3$ and 4.252 billion m$^3$ when YEP is at the speed of −1% and 1%, respectively.

3.4.2. The Changing Trend of Water Requirement in Tianjin

For Tianjin, Tables 12–15 provide the prediction results of water requirement with different growth rates of AVPI, AVSI, AVTI, and YEP in Tianjin. It can be known from Table 12 with the growth rates of AVPI are 1% and 2%, the water requirement will all increase, and the corresponding total water requirement will increase by 31.16 billion m$^3$ and 32.82 billion m$^3$ by 2026, respectively. The decrease is 114 million m$^3$, or the increase is 52 million m$^3$ compared with 2021 respectively.

It can be known from Table 13 as the growth rates of AVSI are −6% and −4%, respectively, the water requirement will all be cut down, and the corresponding total water requirement will change from 11.4 billion m$^3$ to 18.26 billion m$^3$ by 2026. The corresponding change will account for 43.46% and 35.29% of that in 2021, respectively.

From Tables 14 and 15, we can see that the growth rates of AVTI are −2% and 1%, respectively, and the water requirement will all decrease or increase. The corresponding total water requirement is 23.02 billion m$^3$ and 32.78 billion m$^3$ by 2026. The corresponding change accounts for 28.73% and 1.48% of that in 2021, respectively. Similarly, the corresponding total water requirement is 16.11 billion m$^3$ and 42.58 billion m$^3$ when YEP is at the speed of −1% and 1%, respectively.

### 3.4.3. The Changing Trend of Water Requirement in Hebei Province

Judging from the changes in Tables 16–19. It can be seen from Tables 16 and 17 when the growth rates of AVPI are −1% and 1%, respectively, the water requirement will all decrease or increase. The corresponding total water requirement will be 170.8 billion m$^3$ and 195.90 billion m$^3$ by 2026, respectively. The corresponding change will account for 6.1% and 7.69% of that in 2021. Similarly, the corresponding total water requirement is 170.14 billion m$^3$ and 185.91 billion m$^3$ when AVSI at the speed of −1% and 1%, respectively.

Likewise, from Tables 18 and 19, with the growth rates of AVTI are 7% and 9%, respectively, the water requirement will all increase; the corresponding total water requirement will be 261.55 billion m$^3$ and 294.72 billion m$^3$ by 2026. The corresponding change will account for 43.78% and 62.02% of that in 2021. Similarly, the corresponding total water requirement is 196.12 billion m$^3$ and 212.63 billion m$^3$ when YEP is at the speed of 1% and 2%, respectively.

### 3.4.4. The Changing Trend of Water Requirement in Shanxi Province

It can be known from Tables 20–23 when the growth rates of AVPI are −1% and 5%, AVTI is −1% and 5%, and YEP are −1% and 1%, the water requirement will all decrease or increase, the corresponding total water requirement is 6.8 billion m$^3$ and 11.486 billion m$^3$ by 2026 respectively. The corresponding change will account for 6.33% and 58.2% of that in 2021. Similarly, the corresponding total water requirement is 6.766 billion m$^3$ and 13.936 billion m$^3$ when AVSI is at the speed of −1% and 10%, respectively. The corresponding total water requirement is 6.875 billion m$^3$ and 10.119 billion m$^3$ when AVTI is at the speed of −1% and 5%, respectively. The corresponding total water requirement is 4.045 billion m$^3$ and 7.808 billion m$^3$ when YEP is at the speed of −1% and 1%, respectively.

### 3.4.5. The Changing Trend of Water Requirement in Inner Mongolia Autonomous Region

For the Inner Mongolia Autonomous Region, Tables 24–27 present the prediction results of water requirement with different growth rates of AVPI, AVSI, AVTI, and YEP. From Tables 24–27, we can see that with the growth rates of AVPI being −1% and 5%, respectively, the water requirement will all decrease or increase. The corresponding total water requirement is 184.18 billion m$^3$ and 230.64 billion m$^3$ by 2026, respectively. The corresponding change will account for 3.92% and 20.31% of that in 2021. Similarly, the corresponding total water requirement is 180.13 billion m$^3$ and 199.09 billion m$^3$ when YEP at the speed of −1% and 1%, respectively.

It can be known from Table 25 with the growth rates of AVSI being −1% and −2%, respectively, the water requirement will all decrease. The corresponding total water requirement is 174.97 billion m$^3$ and 168.05 billion m$^3$ by 2026. The corresponding change rate accounts for 8.73% and 12.34% of that in 2021, respectively.

For Table 26, as the growth rates of AVTI are 1% and 3%, the water requirement will all increase. The corresponding total water requirement is 225.09 billion m$^3$ and 247.80 billion m$^3$ by 2026. The corresponding change will account for 17.41% and 29.26% of that in 2021, respectively.

## 4. Grey Interval Prediction of Water Requirement in North China

Accurate forecast results can provide valuable reference information for policymakers. Section 3 gives the point prediction results of water requirements in North China through the grey multivariate model. For the purpose of providing more information for policy-

makers, this section further uses the interval grey model to estimate the upper value and lower value of water requirement.

### 4.1. Interval DGMC(1,N) Prediction Model

The accumulation order in grey system theory can represent the priority of different data, which contains different information with different predictive scenarios. Therefore, different accumulation orders can represent different future scenarios. In this section, for the convenience of calculation, the scenarios reflected by the new data are represented by the accumulation order of 0.1, 0.2, 0.3, and 0.4, respectively. The smaller the accumulation order is, the greater the importance of the new data will be. In the scenarios reflected by the new data, there are divergences in different information, and the mean value is used to balance the divergences. The accumulation order 0.6, 0.7, 0.8, and 0.9, respectively, represent the scenarios reflected by the old data. Similarly, in the scenarios reflected by the old data, there are also divergences in different information, and the mean value is also used to balance the divergences. The maximum of two averages is the upper value of the grey interval prediction, and the minimum of two averages is the lower value of the grey interval prediction. Thus, the process modeling of interval DGMC(1,N) can be written as

Step 1: Assume the non-negative sequence $X_1^{(0)} = \left\{ x_1^{(0)}(1), x_1^{(0)}(2), \cdots, x_1^{(0)}(m) \right\}$ as the system characteristic sequence and the corresponding correlation factor sequences are

$$X_2^{(0)} = \left\{ x_1^{(0)}(1), x_1^{(0)}(2), \cdots, x_1^{(0)}(m) \right\}, X_3^{(0)} = \left\{ x_1^{(0)}(1), x_1^{(0)}(2), \cdots, x_1^{(0)}(m) \right\},$$
$$X_N^{(0)} = \left\{ x_1^{(0)}(1), x_1^{(0)}(2), \cdots, x_1^{(0)}(m) \right\}$$

Step 2: For the 0.1-order accumulation sequence, the DGMC(1,N) model is shown in Equation (5)

$$X_1^{(0)}(k) + b_1 z_1^{(0.1)}(k) = \sum_{i=2}^{N} b_1 z_1^{(0.1)}(k) + u \tag{5}$$

$b_1, b_2, \cdots, b_N$ and $u$ are parameters to be estimated. The least squares method is used to minimize the residual sum of squares. The unknown parameters are solved by Equation (6)

$$\left[ \hat{b}_1, \hat{b}_2, \cdots, \hat{b}_u, \hat{u} \right] = \left( B^T B \right)^{-1} B^T Y \tag{6}$$

where,

$$B = \begin{bmatrix} -\frac{x_1^{(0.1)}(1)+x_1^{(0.1)}(2)}{2} & -\frac{x_2^{(0.1)}(1)+x_2^{(0.1)}(2)}{2} & \cdots & -\frac{x_N^{(0.1)}(1)+x_N^{(0.1)}(2)}{2} & 1 \\ -\frac{x_1^{(0.1)}(2)+x_1^{(0.1)}(3)}{2} & -\frac{x_2^{(0.1)}(2)+x_2^{(0.1)}(3)}{2} & \cdots & -\frac{x_N^{(0.1)}(2)+x_N^{(0.1)}(3)}{2} & 1 \\ -\frac{x_1^{(0.1)}(3)+x_1^{(0.1)}(4)}{2} & -\frac{x_2^{(0.1)}(3)+x_2^{(0.1)}(4)}{2} & \cdots & -\frac{x_N^{(0.1)}(3)+x_N^{(0.1)}(4)}{2} & 1 \\ \vdots & \vdots & \vdots & \vdots & \vdots \\ -\frac{x_1^{(0.1)}(m-1)+x_1^{(0.1)}(m)}{2} & -\frac{x_2^{(0.1)}(m-1)+x_2^{(0.1)}(m)}{2} & \cdots & -\frac{x_N^{(0.1)}(m-1)+x_N^{(0.1)}(m)}{2} & 1 \end{bmatrix},$$

$$Y = \begin{bmatrix} x_1^{(0.1)}(2) - x_1^{(0.1)}(1) \\ x_1^{(0.1)}(3) - x_1^{(0.1)}(2) \\ x_1^{(0.1)}(4) - x_1^{(0.1)}(3) \\ \vdots \\ x_1^{(0.1)}(m) - x_1^{(0.1)}(m-1) \end{bmatrix}.$$

Step 3: Assume $\hat{x}_1^{(1)}(1) = x_1^{(0)}(1)$, then

$$\hat{x}_1^{(0.1)}(k) = x_1^{(0)} e^{-\hat{b}_1(k-1)} + \int_1^k e^{-\hat{b}_1(k-1)} f(t)dt. \tag{7}$$

where $f(t) = b_2 x_2^{(0.1)}(t) + b_3 x_3^{(0.1)}(t) + \cdots + b_N x_N^{(0.1)}(t) + u$.

The time response equation gained from the Gaussian formula is

$$\hat{x}_1^{(0.1)}(k) = x_1^{(0)}(k)e^{-\hat{b}_1(k-1)} + \sum_{t=2}^{k} \left\{ e^{-\hat{b}_1\left(k-t+\frac{1}{2}\right)} * \frac{1}{2}[f(t) + f(t-1)] \right\}. \tag{8}$$

Step 4: The fitting sequence is

$$\hat{x}_1^{(0)} = \left\{ \hat{x}_1^{(0)}(1), \hat{x}_1^{(0)}(2), \cdots, \hat{x}_1^{(0)}(m) \right\},$$

where $\hat{x}_1^{(0)}(k) = \hat{x}_1^{(0.1)}(k) - 0.1\hat{x}_1^{(1)}(k-1)$.

Step 5: The model is evaluated by MAPE. If the MAPE is less than 10%, the model is judged to fit well. It is assumed that the correlation factor sequences of the prediction period have been obtained. By the 0.1-order accumulation generator, the accumulated correlation factor sequences are substituted into Equation (8) to obtain the system feature sequence of the prediction period, denoted as $y_{0.1}(m+1), y_{0.1}(m+2), \cdots$.

Step 6: Set the accumulation order to 0.2, 0.3, 0.4, 0.6, 0.7, 0.8, and 0.9, respectively. According to Steps 2–5, the system feature sequences of the prediction period are $y_{0.2}(m+1), y_{0.2}(m+2), \cdots, y_{0.9}(m+1), y_{0.9}(m+2), \cdots$.

Step 7: The interval lower value of the system characteristic sequence at time $1 + m$ is

$$\min\{F_{0.1-0.4}(m+1), F_{0.6-0.9}(m+1)\},$$

The upper interval value at time $1 + m$ is

$$\max\{F_{0.1-0.4}(m+1), F_{0.6-0.9}(m+1)\},$$

where

$$F_{0.1-0.4}(m+1) = \frac{y_{0.1}(m+1) + y_{0.2}(m+1) + y_{0.3}(m+1) + y_{0.4}(m+1)}{4},$$
$$F_{0.6-0.9}(m+1) = \frac{y_{0.6}(m+1) + y_{0.7}(m+1) + y_{0.8}(m+1) + y_{0.9}(m+1)}{4}.$$

Step 8: According to the method of Step 7, the interval of the system characteristic sequence at the other time points is obtained.

In fact, for simple calculation, the accumulation orders are 0.1, 0.2, 0.3, 0.4, 0.6, 0.7, 0.8, and 0.9. The other orders can also be taken theoretically.

### 4.2. Grey Interval Prediction of Water Requirement in Beijing

Taking Beijing's water requirement as the system characteristic sequence and AVPI as the relevant factor sequence, the prediction interval of water requirement is constructed. It can be known from the data that the growth rate of AVPI in Beijing from 2016 to 2021 is $-3\%$. Without considering the influence of AVSI, AVTI, and YEP in Beijing in the future five years, assuming the growth rate of AVPI is $-3\%$. According to the modeling process of the interval DGMC(1,2), the prediction interval of water requirement with the influence of the primary industry in Beijing is given in Figure 2.

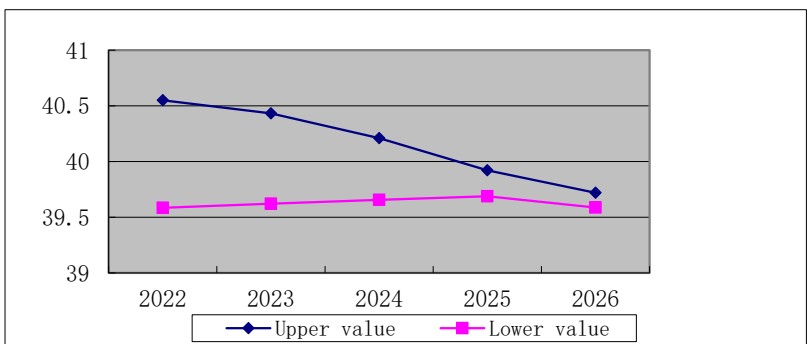

**Figure 2.** The variation interval of water requirement with the influence of primary industry in Beijing.

According to Figure 2, with the growth rate of AVPI as the average annual growth rate in the previous five years, the fluctuation of prediction water requirement in Beijing will be first large and then small. In fact, AVPI exhibits a downward trend year by year. The development of agricultural science and technology is an irresistible trend in the future. Thus, the demand for water in the primary industry will also tend to be stable. It indicates that the predicted results are rational.

Likewise, interval DGMC(1,2) is established by taking AVSI, AVTI, and YEP of Beijing as the relevant factors, respectively. Assuming that the future growth rate of each relevant factor is the average growth rate in the previous five years. That is, under the condition of a fixed growth rate, the water requirement under the influence of each relevant factor in Beijing is predicted. The interval prediction results are given in Figures 3–5.

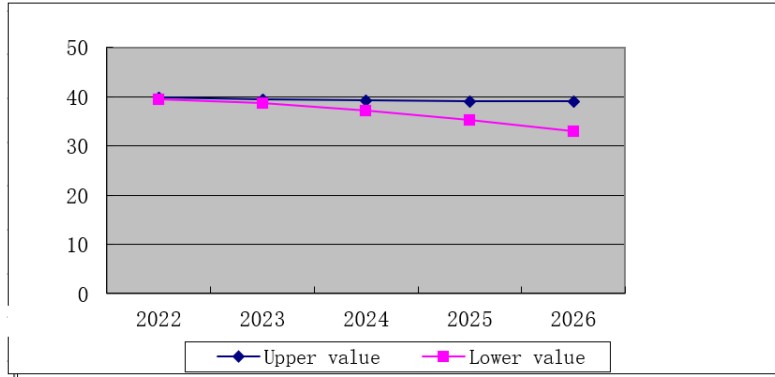

**Figure 3.** The variation interval of water requirement with the influence of secondary industry in Beijing.

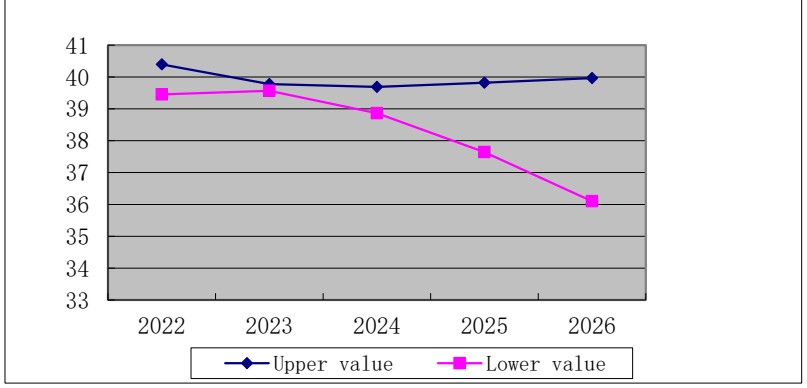

**Figure 4.** The variation interval of water requirement with the influence of tertiary industry in Beijing.

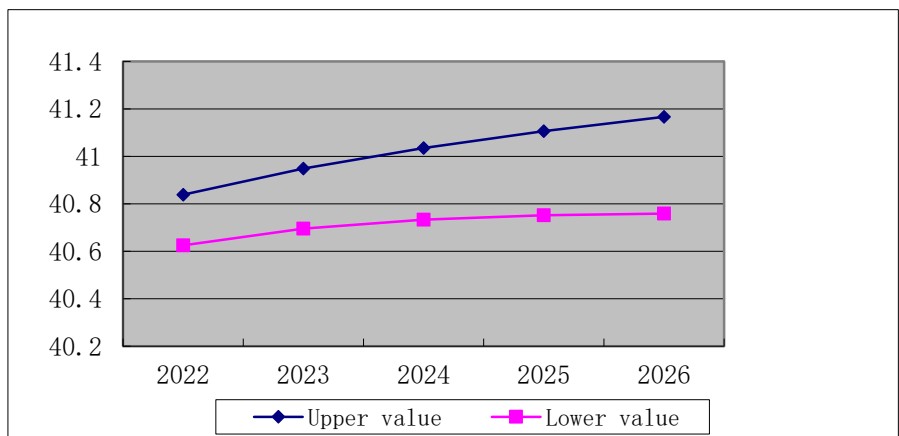

**Figure 5.** The variation interval of water requirement with the influence of population in Beijing.

It can be seen from Figures 3–5 that the variation range of water requirement with the influence of the secondary industry in Beijing will narrow down slightly with the growth rate is the average annual growth rate in the previous five years. The predicted interval of water requirement with the influence of the tertiary industry will fluctuate greatly or widen gradually under the influence of the population. This indicates that the influence of the secondary industry on water requirement shows little change, and the influence of the tertiary industry and population on water requirement will alter greatly. From the perspective of water conservation, attention should be paid to tertiary industry development and improving the public's awareness of water conservation in order to reduce the impact of population.

### 4.3. Grey Interval Prediction of Water Requirement in Tianjin

Taking Tianjin's water requirement as the system characteristic sequence and AVPI as the relevant factor sequence, the calculative process is consistent with Beijing. The prediction interval of water requirement under the influence of the primary industry in Tianjin is given in Figure 6. It indicates that the fluctuation of the upper value and the lower value of Tianjin's water requirement will show an upturn when the growth rate of AVPI continues the average annual growth rate in the previous five years. It means that the increase in Tianjin's AVPI depends on the increase in water requirement.

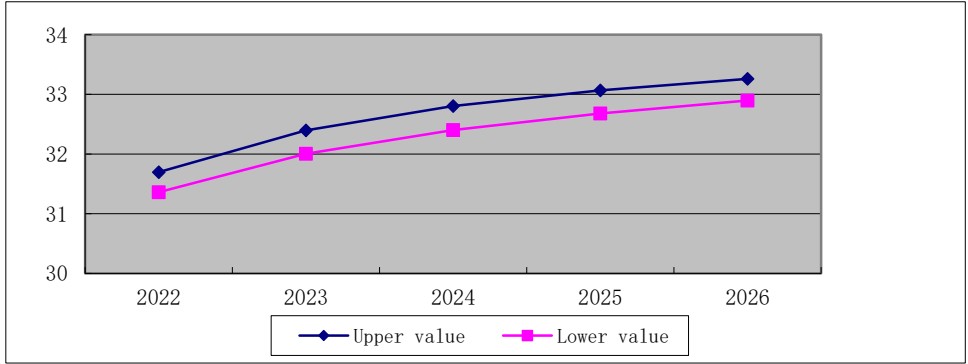

**Figure 6.** The variation interval of water requirement with the influence of primary industry in Tianjin.

Likewise, the interval DGMC(1,2) model is established by taking AVSI, AVTI, and YEP of Tianjin as the relevant factors, respectively. The calculative process is consistent with Beijing. The interval prediction results are given in Figures 7–9.

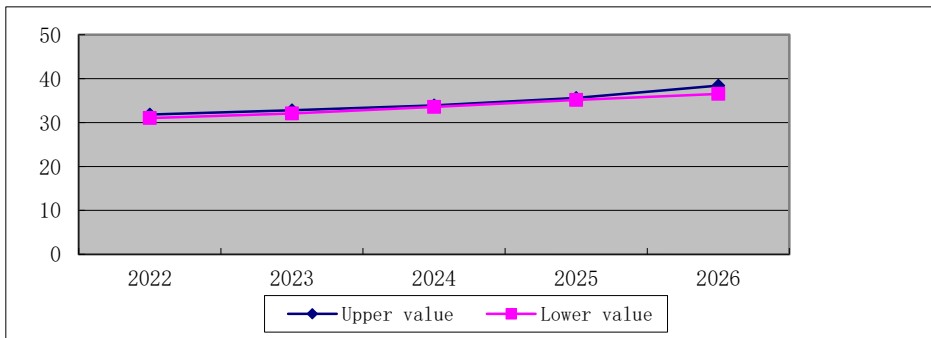

**Figure 7.** The variation interval of water requirement with the influence of the secondary industry in Tianjin.

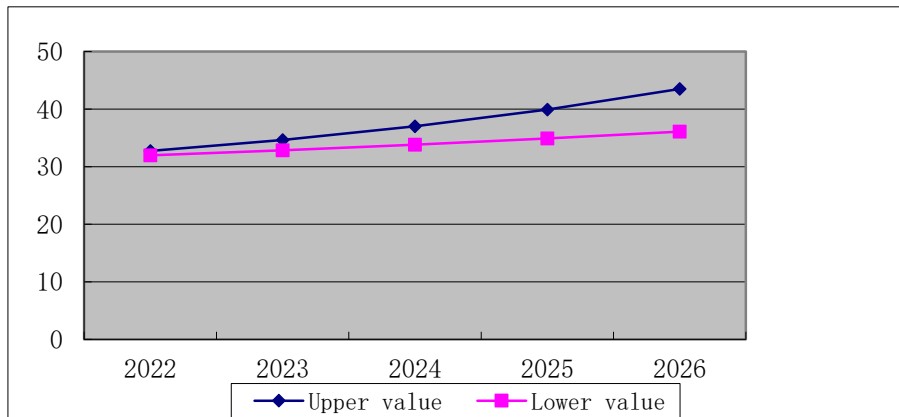

**Figure 8.** The variation interval of water requirement with the influence of the tertiary industry in Tianjin.

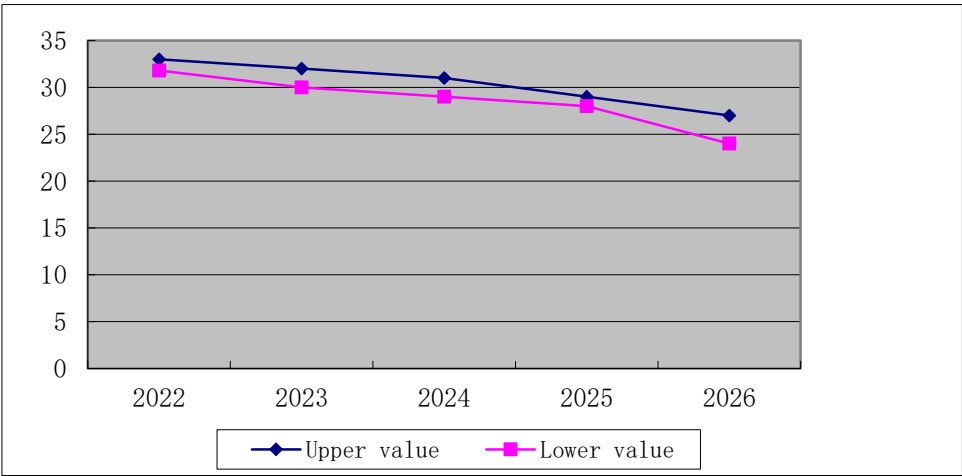

**Figure 9.** The variation interval of water requirement with the influence of the population in Tianjin.

Figure 7 shows that the fluctuation range of water requirement in Tianjin will be quite narrow, with the growth rate of AVSI being the average annual growth rate in the past five years. Figure 8 shows that the fluctuation range of water requirement in Tianjin will broaden with the growth rate of AVTI is the average annual growth rate in the past five years. Figure 9 indicates that the fluctuation of the upper value and the lower value of water requirement in Tianjin will decline if the growth rate of YEP is the average annual growth rate in the previous five years.

### 4.4. Grey Interval Prediction of Water Requirement in Hebei Province

Taking Hebei's water requirement as the system characteristic sequence and AVPI as the relevant factor sequence, the calculative process is consistent with Beijing. The prediction interval of water requirement with the influence of the primary industry in Hebei is shown in Figure 10. It indicates that both the upper value and the lower value of Hebei's water requirement will exhibit a slight downward trend, with the growth rate of AVPI being the average annual growth rate in the past five years. This is because the impact on the total water requirement is also slightly decreasing (The proportion of the primary industry in the gross domestic product of Hebei is decreasing slightly).

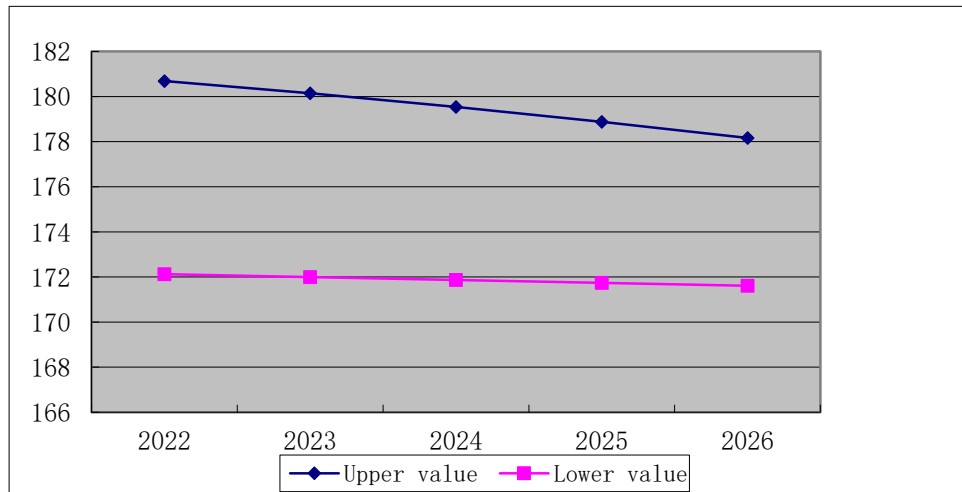

**Figure 10.** The variation interval of water requirement with the influence of primary industry in Hebei.

Likewise, the interval DGMC(1,2) model is established by taking AVSI, AVTI, and YEP of Hebei as the relevant factors, respectively. The calculative process is consistent with Beijing. The interval prediction results are shown in Figures 11–13.

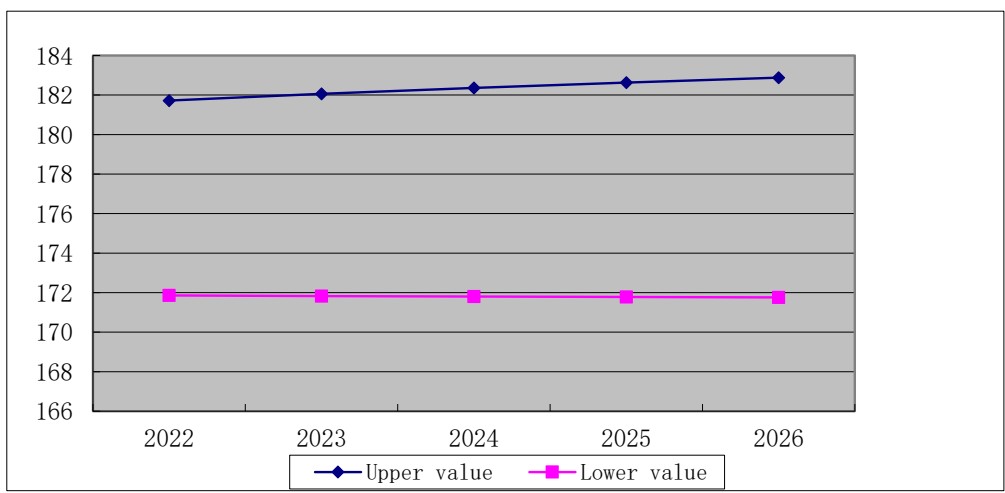

**Figure 11.** The variation interval of water requirement with the influence of the secondary industry in Hebei.

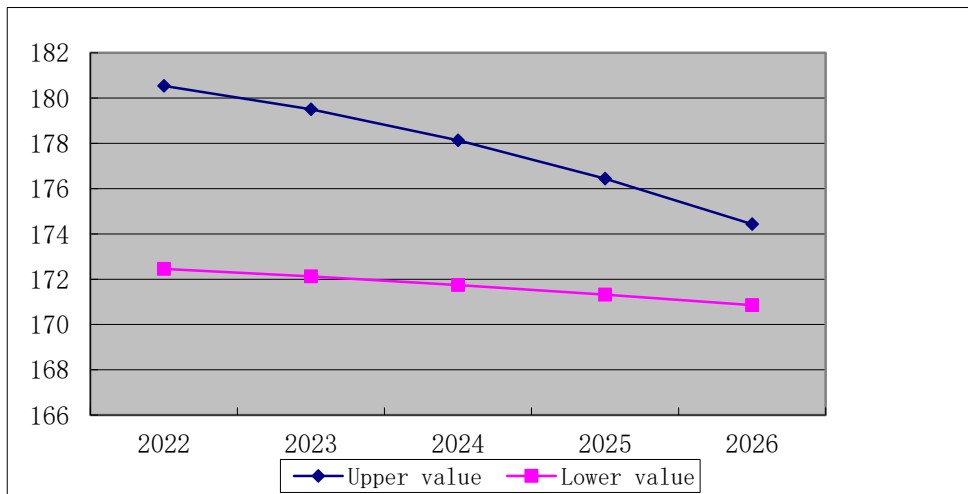

**Figure 12.** The variation interval of water requirement with the influence of the tertiary industry in Hebei.

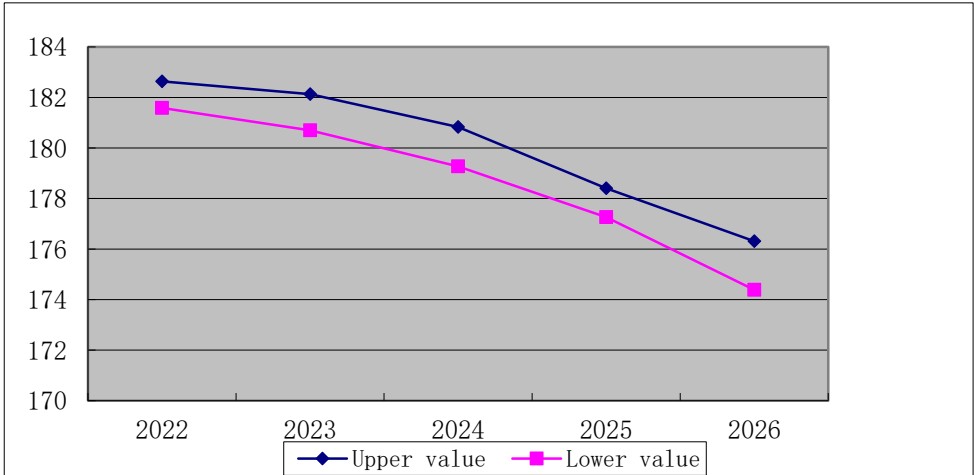

**Figure 13.** The variation interval of water requirement with the influence of population in Hebei Province.

As shown in Figure 11, as the growth rate of AVSI is the average annual growth rate in the past five years, the upper value of water requirement in Hebei will increase slightly, and the lower value will remain nearly unchanged. As for Figure 12, with the growth rate of AVTI as the average annual growth rate in the previous five years, the fluctuation range of water requirement in Hebei will be narrow. For Figure 13, with the growth rate of YEP as the average annual growth rate in the past five years, both the fluctuation of the upper value and the lower value of water requirement in Hebei will show a downward trend. It is because YEP in Hebei has changed very little from 2016 to 2021, and the growth rate is slow and even negative in some years. It means that the decline of population in the next five years will lead to a decrease in the impact of population on water requirements.

### 4.5. Grey Interval Prediction of Water Requirement in Shanxi Province

Taking Shanxi's water requirement as the system characteristic sequence and AVPI as the relevant factor sequence, the calculative process is consistent with Beijing. The predicted interval of water requirement with the influence of the primary industry in Shanxi is shown in Figure 14. It indicates that both the upper value and the lower value of Shanxi's water requirement will show a downward trend, while the growth rate of AVPI is the average annual growth rate in the previous five years.

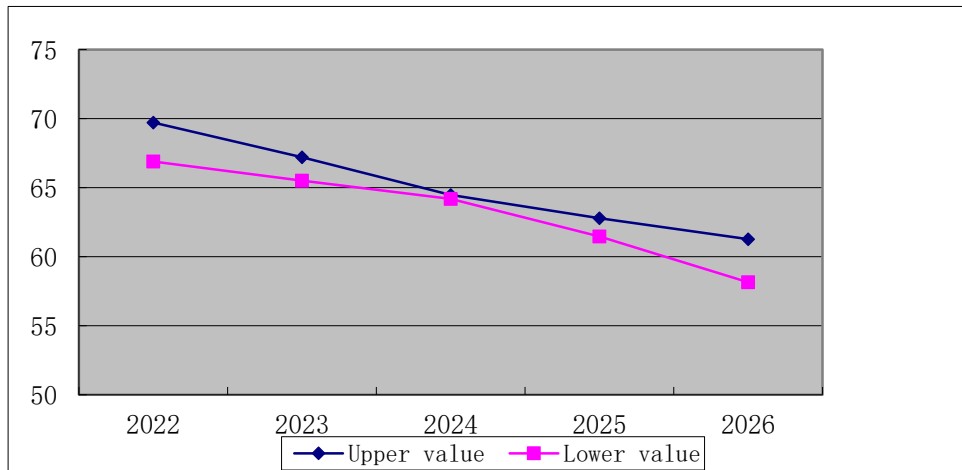

**Figure 14.** The variation interval of water requirement with the influence of primary industry in Shanxi Province.

Likewise, the interval DGMC(1,2) model is established by taking AVSI, AVTI, and YEP of Shanxi as the relevant factors, respectively. The calculative process is consistent with Beijing. The interval prediction results are shown in Figures 15–17.

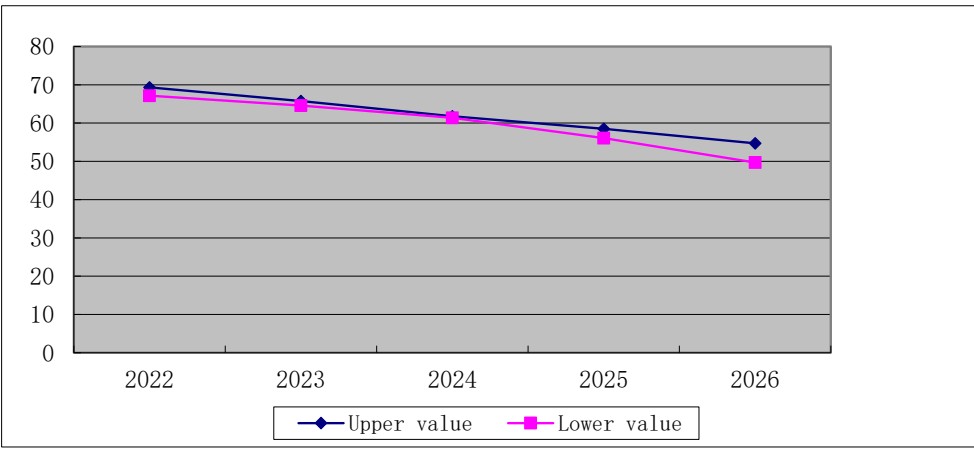

**Figure 15.** The variation interval of water requirement with the influence of secondary industry in Shanxi Province.

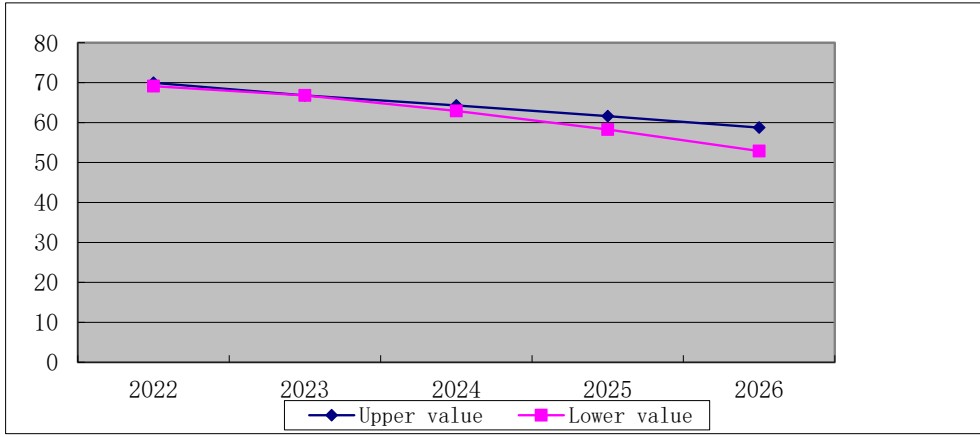

**Figure 16.** The variation interval of water requirement with the influence of the tertiary industry in Shanxi Province.

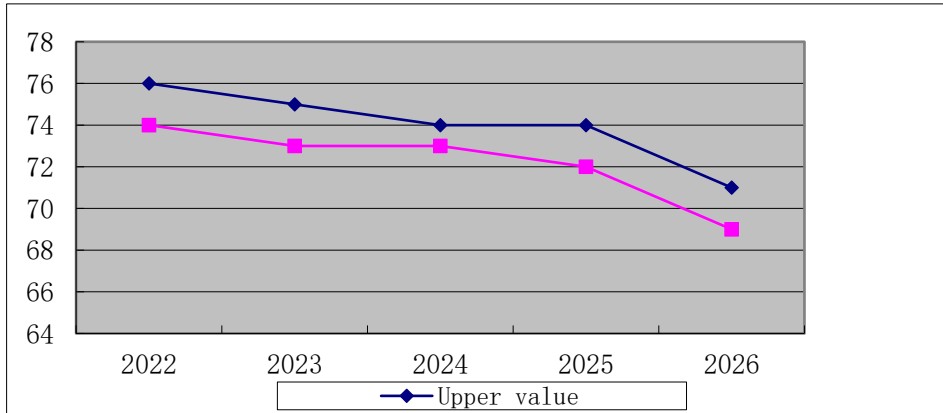

**Figure 17.** The variation interval of water requirement with the influence of population in Shanxi Province.

Figure 15 demonstrates that the fluctuation of the upper value and the lower value of Shanxi's water requirement will show a downward trend, with the growth rate of AVSI being the average annual growth rate in the previous five years. Figure 16 indicates that the fluctuation of the upper value and the lower value of Shanxi's water requirement will decrease with the growth rate of AVTI is the average annual growth rate in the past five years. Figure 17 reveals that the fluctuation of the upper value and the lower value of water requirement in Shanxi will decline if the growth rate of YEP is the average annual growth rate in the previous five years.

*4.6. Grey Interval Prediction of Water Requirement in Inner Mongolia*

Taking Inner Mongolia's water requirement as the system characteristic sequence and AVPI as the relevant factor sequence, the calculative process is consistent with Beijing. The prediction interval of water requirement with the influence of the primary industry in Inner Mongolia is presented in Figure 18. According to Figure 18, on the premise that the growth rate of AVPI is the average annual growth rate in the previous five years, the fluctuation range of water requirement in Inner Mongolia will first narrow and then broaden.

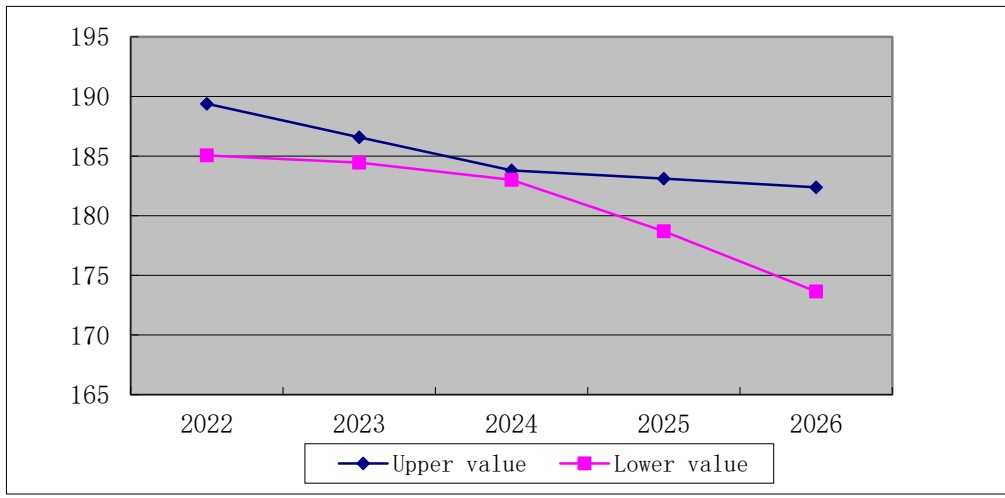

**Figure 18.** The variation interval of water requirement with the influence of the primary industry in Inner Mongolia.

Likewise, the interval DGMC(1,2) model is established by taking AVSI, AVTI, and YEP of Inner Mongolia as the relevant factors, respectively. The calculative process is consistent with Beijing. The interval prediction results are shown in Figures 19–21.

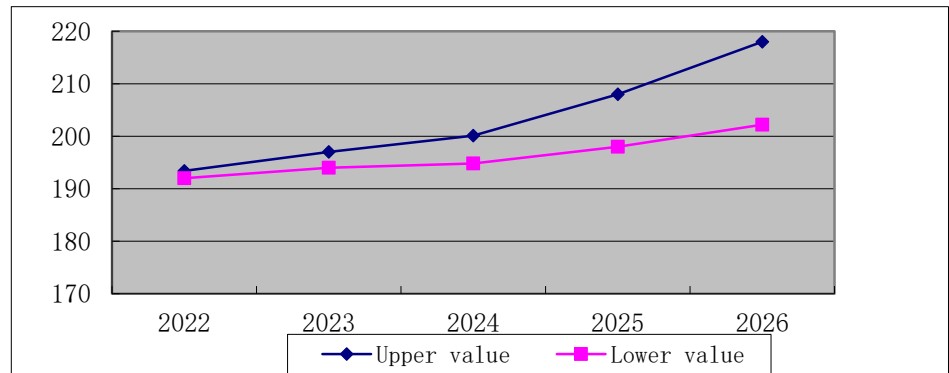

**Figure 19.** The variation interval of water requirement with the influence of the secondary industry in Inner Mongolia.

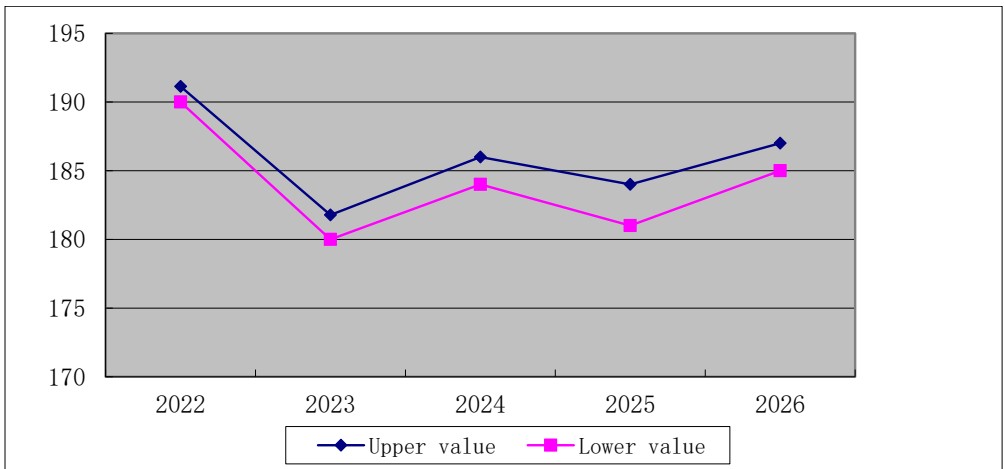

**Figure 20.** The variation interval of water requirement with the influence of the tertiary industry in Inner Mongolia.

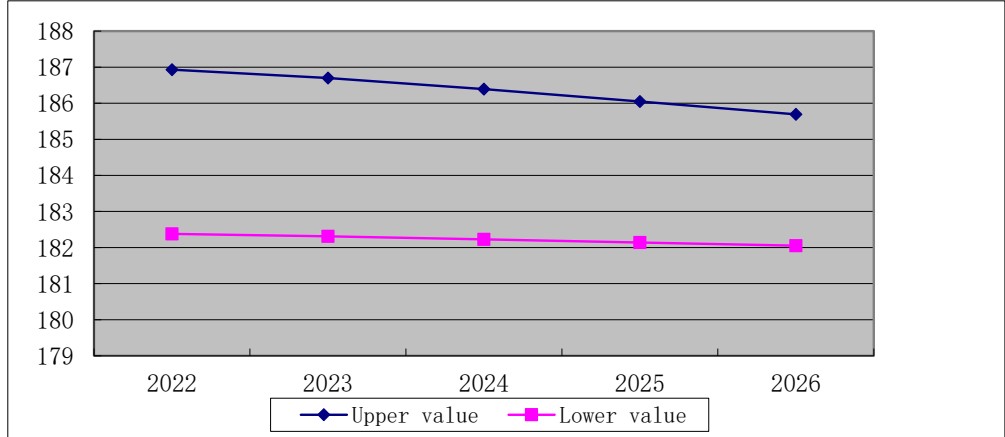

**Figure 21.** The variation interval of water requirement with the influence of population in Inner Mongolia.

As for Figure 19, as the growth rate of AVSI is the average annual growth rate in the previous five years, the fluctuation range of water requirements in Inner Mongolia will widen year by year. Figure 20 shows that with the growth rate of AVTI as the average annual growth rate in the previous five years, the fluctuation range of water requirement in Inner Mongolia will change greatly and remain in an unstable status. According to

Figure 21, with the growth rate of YEP being the average annual growth rate in the past five years, both the upper value and the lower value of water requirement in Inner Mongolia will decrease slightly.

## 5. Conclusions and Discussion

This study selects the variable accumulation grey multivariable mode to predict the water requirement in North China, which ensures the prediction accuracy of water requirement and lays a foundation for the accurate analysis of the water requirement in the region in the later period. According to the research and prediction analysis of the above models, this paper draws the following conclusions.

AVPI, AVSI, AVTI, and YEP are selected as the four influence factors. Under different growth scenarios, the DGMC(1,2) model is constructed to predict the water requirement of five provinces in North China by using these four factors. The forecasting results of water requirements from 2022 to 2026 under the relevant indicators of each province are obtained. The water requirement will increase if the AVPI increases in Beijing, Tianjin, Hebei, Shanxi, and Inner Mongolia, respectively. The water requirement will increase if the AVSI increases in Beijing, Tianjin, Hebei, Shanxi, and Inner Mongolia, respectively. The water requirement will increase if the AVTI increases in Tianjin, Hebei, and Shanxi, respectively. The water requirement has an increasing trend if the AVTI increases in Beijing and Inner Mongolia, respectively. The water requirement will increase if the YEP increases in Beijing, Tianjin, Hebei, Shanxi, and Inner Mongolia, respectively. Especially in Tianjin, the water requirement will increase at a higher speed if the YEP increases at a low speed. According to the prediction results, the variation trend of water requirement in each region was analyzed in detail, and the corresponding suggestions were put forward.

The interval DGMC(1,$N$) model is proposed to forecast the water requirement of the five provinces in North China. Assuming that the future growth rate of relevant factors is the same as that of the past five years. Under a fixed growth rate, the water requirement under the influence of each relevant factor is predicted. According to the forecasting results, although the water requirement can control the state gradually, it is still essential to monitor the development of related water-intensive industries in order to save water.

In this paper, there are still some problems that need to be further discussed. Although the socioeconomic indicators are selected to predict the water requirement, the climate factors and the influence of government water conservancy projects on the water requirement are not taken into account. More influence factors will be selected for comprehensive prediction in other areas in the future so as to grasp the changing trend of water requirements. In addition, due to the less available data, we currently use the grey model to obtain the corresponding results. In the prospective study, a large amount of data will be collected to build a statistical prediction model.

**Author Contributions:** Conceptualization, L.W.; methodology, L.W.; software, L.W.; validation, L.W.; formal analysis, L.W.; investigation, L.W.; resources, L.W.; data curation, L.W.; writing—original draft preparation, Y.M.; writing—review and editing, Y.M.; visualization, Y.M.; supervision, L.W.; project administration, L.W.; funding acquisition, Y.M. All authors have read and agreed to the published version of the manuscript.

**Funding:** The relevant research are supported by the National Natural Science Foundation of China (71871084, U20A20316), Graduate Demonstration Course in Hebei Province (KCJSX2022095), the key research project in humanity and social science of Hebei Education Department (ZD202211), the Natural Science Foundation of Hebei Province (E2020402074) and the Social Science Federation Project of Handan (2023072, 2023088).

**Data Availability Statement:** All data included in this study are available upon request by contact with the corresponding author.

**Conflicts of Interest:** The authors declare no conflict of interest.

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
