# Peer review of "Water Requirement in North China from Grey Point Prediction and Grey Interval Prediction"

_water, doi:10.3390/w15081453_

Round 1

Reviewer 1 Report

Tables 2 to 6 do not include the unit of measurements of the water requirements.

The paper present the application of a method to forecast the water requirement but there is no way to evaluate how good are the results. No comparison with other methods has been provided, no comparison was shown for 2022, a comparison between predictive values and real values should be made for 2022. 

How can you be sure, based on what you have done, that the method applied ensures the prediction accuracy of water requirement as you were mentioning in your conclusions. 

Author Response

Detailed Response to Reviewer 1

We thank very much for the constructive and helpful comments from the anonymous reviewer. We have made numerous changes based on suggestions. The followings are the detailed responses to the comments and suggestions. Line numbers used in the reply section refer to text in the revised manuscript.

  1. Tables 2 to 6 do not include the unit of measurements of the water requirements.

It has been corrected in Table 1.

Table 1. Social and economic index

Index.

Unit

AVPI

100 million Yuan

AVSI

100 million Yuan

AVTI

100 million Yuan

YEP

10,000 people

Water requirement

100 million m3

  1. The paper present the application of a method to forecast the water requirement but there is no way to evaluate how good are the results. No comparison with other methods has been provided, no comparison was shown for 2022, a comparison between predictive values and real values should be made for 2022. 

Due to impact of 19-COVID, the real values for 2022 are not released by now. The forecasting results of the water requirement of five provinces in North China are consistent with the expert judgment. In the future, we will compare predictive values with real values for 2022 when the data is released. 

The model is evaluated by mean absolute percentage error (MAPE= ). If the MAPE is less than 10%, the model is judged to fit well.

The comparison data between the fitted value and the real value is presented in Table 7. The MAPE of DGMC(1,2) is 1.04%, far less than 10%. Its error is small. It indicates that the model can obtain better predictive result of the water requirement in Beijing.

  1. How can you be sure, based on what you have done, that the method applied ensures the prediction accuracy of water requirement as you were mentioning in your conclusions. 

The model is evaluated by mean absolute percentage error (MAPE= ). If the MAPE is less than 10%, the model is judged to fit well.

The DGMC(1,2) is used for predicting the water requirement. Taking AVPI in Beijing as an example, we will provide the modeling process of DGMC(1,2). AVPI, , is the influence factor of water requirement. In order to obtain the minimum MAPE, the optimal order is solved by the particle swarm optimization algorithm.

When

The parameters of DGMC(1,2) can be obtained by the least squares method.

Therefore, the time response formula of the DGMC(1, 2) is

Where .

Thus .

The comparison data between the fitted value and the real value is presented in Table 7. The MAPE of DGMC(1,2) is 1.04%, far less than 10%. Its error is small. It indicates that the model can obtain better predictive result of the water requirement in Beijing.

Table 7. The fitted value and the real value of water requirement.

Year

The real value

The fitted value

2016

38.8

38.8

2017

39.5

39.05

2018

39.3

40.017

2019

41.7

40.67

2020

40.6

40.68

2021

40.8

40.53

Step 5: The model is evaluated by MAPE. If the MAPE is less than 10%, the model is judged to fit well. It is assumed that the correlation factor sequences of the prediction period have been obtained. By the 0.1-order accumulation generator, the accumulated correlation factor sequences are substituted into the Eq.(8) to obtain the system feature sequence of the prediction period, denoted as .

Reviewer 2 Report

This paper deals with forecasting the water demand of five provinces in North China using a method based on the grey systems theory. The article has a good structure. The subject is significant and important. The results can have practical value and be used for policymaking. But some flaws must be fixed:
1. Why does the article not have a literature review section? This is not acceptable. In reviewing the literature, the authors should look at the previous research in predicting the water demand of different regions and the methods used, as well as the methods they use, that is, the methods based on the theory of grey systems. It is recommended to use review articles in the field of grey systems theory, especially review articles in the field of applications of grey systems theory in the field of economic and social systems or sustainability, because this can help the readers get appropriate information about the background of the subject in a short time.
2. I think the article's abstract does not provide enough information and should be improved. The importance of the issue, solution method, and results should be mentioned.
3. The authors have used the "Interval DGMC(1,N) Prediction Model" method. This is a suitable version, but why did they not mention the reason for choosing this method? What is the advantage of this method compared to other econometric methods?
4. Why does this article not have a results and discussion section? How can the authors present the output results without adequate analysis? This is not acceptable.

Author Response

Detailed Response to Reviewer 2

We thank very much for the constructive and helpful comments from the anonymous reviewer. We have made numerous changes based on suggestions. The followings are the detailed responses to the comments and suggestions. Line numbers used in the reply section refer to text in the revised manuscript.

This paper deals with forecasting the water demand of five provinces in North China using a method based on the grey systems theory. The article has a good structure. The subject is significant and important. The results can have practical value and be used for policymaking. But some flaws must be fixed:

  1. Why does the article not have a literature review section? This is not acceptable. In reviewing the literature, the authors should look at the previous research in predicting the water demand of different regions and the methods used, as well as the methods they use, that is, the methods based on the theory of grey systems. It is recommended to use review articles in the field of grey systems theory, especially review articles in the field of applications of grey systems theory in the field of economic and social systems or sustainability, because this can help the readers get appropriate information about the background of the subject in a short time.

Thanks for your suggestion, and we have added some sentences accordingly. Xu et al. used fractional-order cumulative discrete grey model to forecast agricultural water demand in 2 regions of China, the results indicate that the model has superior prediction property than the grey model with one variable [27]. Wu et al. established a new grey water requirement prediction model, explored the parameter estimation and error testing methods on it, and used this model to predict the water consumption in Chongqing [28]. Besides, Qiao et al. proposed fractional cumulative grey forecasting model (FGM(1,1)) to predict the water demand in various regions of China [29].

  1. I think the article's abstract does not provide enough information and should be improved. The importance of the issue, solution method, and results should be mentioned.

Thanks for your suggestion, and we have added some sentences accordingly.

  1. The authors have used the "Interval DGMC(1,N) Prediction Model" method. This is a suitable version, but why did they not mention the reason for choosing this method? What is the advantage of this method compared to other econometric methods?

Traditional econometric forecasting methods on water requirement tend to require more data and do not value new data, so the prediction effect is not good. Grey multivariable model not only requires less data and attaches importance to new data, but also can make more accurate judgment on complex system and provide valuable information for decision maker. The cost of grey multivariable model is small. Therefore we choose grey multivariable model.

  1. Why does this article not have a results and discussion section? How can the authors present the output results without adequate analysis? This is not acceptable.

Conclusion and discussion

This study selects the variable accumulation grey multivariable mode to predict the water requirement in North China, which ensures the prediction accuracy of water requirement and lays a foundation for the accurate analysis of the water requirement in thei region in the later period. According to the research and prediction analysis of the above models, this paper draws the following conclusions.

AVPI, AVSI, AVTI and YEP are selected as four influence factors. Under different growth scenarios, the DGMC(1,2) model is constructed to predict the water requirement of five provinces in North China by using these four factors. The forecasting results of water requirement from 2022 to 2026 under the relevant indicators of each province are obtained. The water requirement will increase if the AVPI will increase in Beijing, Tianjin, Hebei, Shanxi and Inner Mongolia respectively. The water requirement will increase if the AVSI will increase in Beijing, Tianjin, Hebei, Shanxi and Inner Mongolia respectively. The water requirement will increase if the AVTI will increase in Tianjin, Hebei and Shanxi respectively. The water requirement has an increase trend if the AVTI will increase in Beijing and Inner Mongolia respectively. The water requirement will increase if the YEP will increase in Beijing, Tianjin, Hebei, Shanxi and Inner Mongolia respectively. Especially in Tianjin, the water requirement will increase at a higher speed if the YEP will increase at a low speed. According to the prediction results, the variation trend of water requirement in each region was analyzed in detail and the corresponding suggestions is put forward.

The interval DGMC(1,N) model is proposed to forecast the water requirement of the five provinces in North China. Assuming that the future growth rate of relevant factors is the same as that of the past five years. Under a fixed growth rate, the water requirement under the influence of each relevant factors is predicted. According to the forecasting results, although the water requirement can control the state gradually, it is still essential to monitor the development of related water-intensive industries in order to save water.

In this paper, there are still some problems that need to be further discussed. Although the socioeconomic indicators are selected to predict the water requirement, the climate factors and the influence of government water conservancy projects on the water requirement are not taken into account. More influence factors will be selected for comprehensive prediction in other areas in the future so as to grasp the change trend of water requirement. In addition, due to the less available data, we currently use the grey model to obtain the corresponding results. In the prospective study, a large amount of data will be collected to build a statistical prediction model.

Round 2

Reviewer 2 Report

With kindest regards, I think the paper can be published in the current version.

Author Response

  1. English is improved.
  2. Validating the model of water requirement in Beijing is added.
  3. All references are relevant to the contents of the manuscript.